# Thyroid-Stimulating Hormone Stimulation Tests in the Bottlenose Dolphin (*Tursiops truncatus*)

**Dorian S. Houser** [1,*] , **Cory Champagne** [1] **and Daniel E. Crocker** [2]

1   National Marine Mammal Foundation, 2240 Shelter Island Drive, Suite 200, San Diego, CA 92106, USA;
    cory.champagn@gmail.com
2   Department of Biology, Sonoma State University, 1801 East Cotati Avenue, Rohnert Park, CA 94928, USA;
    crocker@ssu.edu
*   Correspondence: dorian.houser@nmmf.org; Tel.: +1-877-360-5522 (ext. 112)

**Abstract:** Stimulation of the thyroid with thyroid-stimulating hormone (TSH) is a potentially useful diagnostic of thyroid dysfunction, but little is known about the response of the thyroid to TSH stimulation in bottlenose dolphins (*Tursiops truncatus*). To better characterize the response of the dolphin thyroid to TSH stimulation, five adult dolphins participated in a TSH stimulation study. Dolphins voluntarily beached onto a padded mat and were given a 1.5 mg intramuscular injection of human recombinant TSH. Blood samples collected the day prior, at multiple intervals the day of, and daily for three days after the injection were analyzed via radioimmunoassay for free and total triiodothyronine (fT3 and tT3), and free and total thyroxine (fT4 and tT4). Significant increases in circulating fT3, fT4, and tT4 were observed with peaks occurring for all hormones the day after the TSH injection; maximal increases were 44%, 47%, and 23% for each hormone, respectively. Temporal patterns in the hormones potentially reflected feedback mechanisms countering the surge in fT3 following stimulation. Though recombinant human TSH was effective at stimulating hormone release, it is likely that use of dolphin or dolphin-derived TSH would enhance the clinical utility of the stimulation test, as would the development of antibodies specific to dolphin TSH.

**Keywords:** thyroid stimulation; thyroxine; triiodothyronine; cetacean

## 1. Introduction

Proper thyroid function and thyroid hormone regulation are critical for organismal health as thyroid hormones affect nearly every organ system of the body. Thyroid-stimulating hormone (TSH), or thyrotropin, is the primary hormone stimulating thyroid hormone released from the thyroid gland. Produced by the anterior pituitary in response to thyrotropin-releasing hormone (TRH), TSH stimulates the release of triiodothyronine (T3) and thyroxine (T4) from thyroid follicular cells. The metabolically active T3 is primarily responsible for regulation of growth, development, and metabolism through genomic and non-genomic effects on numerous biochemical pathways. T3 is metabolically active in its free (unbound) form and the availability of free T3 is regulated both by its rate of conversion from the metabolically less active T4 and the degree to which it is bound to the proteins thyroxine-binding globulin, transthyretin, and albumin. In addition, regulation can be achieved by converting T4 into reverse T3 (rT3), which is commonly accepted as an end product of T4 degradation and which can weakly compete with T3 in binding to T3 thyroid nuclear hormone receptors. Given the importance of thyroid hormone to organismal health, the study of thyroid hormone production and regulation has received extensive attention within the field of human medicine.

The bottlenose dolphin (*Tursiops truncatus*) is a fully-aquatic mammal that has undergone dramatic physiological adaptations to the thermoregulatory and diving challenges of an aquatic existence. It is also the most commonly held cetacean at marine parks, zoos, and aquaria, establishing a need to understand its normal physiological function for purposes

of veterinary care and animal welfare. Thyroid hormones in bottlenose dolphins presumably support the same metabolic functions as observed in humans and other terrestrial mammals, yet there are distinct differences that suggest possible alterations in metabolic regulation and functionality that require understanding in order to facilitate the diagnosis and treatment of thyroid-based metabolic disease. Most notably, rT3 in dolphins (and other odontocetes) occurs at levels that far exceed those observed in humans and other terrestrial mammals [1–3], although the reason for the high rT3 levels has not yet been determined. Levels of total thyroxine (tT4), the sum of free and protein-bound forms, also commonly fall within the high-end of the normal range for humans: >90 nM, [4,5]. Given these distinctions, there remains little information on the normal variation of thyroid hormones in dolphins and how they are affected by intrinsic (e.g., sex, age, reproductive status) and extrinsic (e.g., seasons, temperature, light cycles) factors. To date, thyroid hormone measurements in bottlenose dolphins have come from wild-caught dolphins [2,6] and those held under human care [2,7], yet differences in potential environmental factors and intrinsic factors make it difficult to confidently establish thyroid hormone levels and patterns for clinical use.

One approach to assessing thyroid function is through stimulation of the hypothalamic-pituitary-thyroid axis, either through the administration of TRH to stimulate TSH release from the anterior pituitary, or administration of TSH to directly stimulate the production and release of T3 and T4 from the thyroid gland. TSH stimulation tests have been used in belugas (*Delphinapterus leucas*) and dolphins [1,7,8]. These studies were limited to one or a few subjects and, in some cases, were performed in wild-caught animals with potentially unknown confounding variables (e.g., underlying health conditions, effects of handling stress). The study reported here builds on a prior TSH stimulation test conducted in a bottlenose dolphin [7] in order to advance the understanding of dolphin thyroid function and the subsequent dynamics of circulating thyroid hormones post-stimulation. The study increases the sample size of subjects to bolster confidence in the observed hormonal response in hopes of advancing the procedure as a clinical tool in the veterinary care of the bottlenose dolphin.

## 2. Materials and Methods

### 2.1. Study Animals

Five adult bottlenose dolphins participated in the TSH stimulation study (Table 1). Study animals were housed in open-water netted enclosures within San Diego Bay, CA, and were maintained by the United States Navy Marine Mammal Program (MMP). Study animals were fed a breakout of fish commensurate with veterinarian prescribed diets. All dolphins were fed a mix of herring (*Clupea harengus*), mackerel (*Scomber scombrus*), squid (*Doryteuthis opalescens*), and capelin (*Mallotus villosus*) throughout the study period. Samples were collected from March to June to minimize potential seasonal variations in thyroid responsiveness to the stimulation.

**Table 1.** Identification code, age, and sex for individual dolphins participating in the thyroid stimulation.

| Animal ID | Mass (kg) | Sex | Age |
| --- | --- | --- | --- |
| TRO | 210 | M | 22 |
| TYH | 204 | M | 33 |
| COL | 221 | M | 13 |
| OLY | 195 | M | 30 |
| BLU | 201 | F | 49 |

### 2.2. Experimental Design

Voluntary blood samples were collected daily for one day before and three days after performance of a thyroid stimulation (TS) test. On these days, dolphins voluntarily participated in blood sampling by presenting the ventral surface of their fluke for blood sampling in exchange for a fish reward. Samples were collected from the arteriovenous

plexus using a 21G, 1.25″ winged sampling needle. Samples were collected into either chilled serum or EDTA plasma vacutainers (BD & Co., Franklin, NJ, USA). Voluntary blood collections were conducted at ~09:00, three hours after receiving a 0.5 kg fish meal (i.e., after sufficient time to reach a post-absorptive state [9]). Blood samples were immediately carried to an on-site facility, centrifuged at 1090 *g* for 15 min, and the serum collected and stored at −80 °C until subsequent analysis.

Dolphins were fed a 0.5 kg fish meal at 06:00 on the day of the TS test. Dolphins voluntarily beached onto a padded mat located under a covered hut three hours after the feeding. The dolphin then remained out of water for an hour while repeated blood samples were collected. (Maintaining the dolphins out of the water for an hour after the injection was implemented as a safety precaution; the procedure had not previously been performed at the MMP, so access to the dolphins was maintained in case rapid veterinary access was needed.) Just after beaching, the initial blood sample was rapidly collected. Blood samples were collected by venipuncture of the peduncle vessel using an 18 G, 1.5″ blood collection needle. The time of the blood collection was considered as time = 0. Immediately after the initial blood collection, the dolphin received a 1.5 mg dose (7–8 µg/kg) of a recombinant human TSH (Thyrogen; Genzyme Corporation, Cambridge, MA, USA) via intramuscular injection into the epaxial muscle. The initial venipuncture in the dolphin's peduncle was maintained for the duration of the test to permit repeated blood sampling every 15 min over the hour. Throughout the TS test, dolphins were kept wet by applying seawater to their skin with either wet towels or sprayers. After an hour, dolphins were returned to the water and resumed a normal feeding schedule. Voluntary blood samples continued to be collected at two and four hours after the TSH injection and then daily for the three days following the test. Initial processing of blood samples was performed as described above.

### 2.3. Hormone Assays

Free T3 (fT3), free T4 (fT4), total T3 (tT3), and tT4 were measured in duplicate from serum using antibody-coated tube RIA kits (Siemens, Inc., Los Angeles, CA, USA; since discontinued). Average replicate CVs were 2.3, 2.2, 1.8, and 2.4% for fT3, tT3, fT4, and tT4, respectively. Each of the kits was previously validated for use in the bottlenose dolphin [10]. Reverse T3 was measured in duplicate from serum using an EIA kit (Alpco Inc. cat# 38-RT3H-R125). Because dolphin rT3 is at levels that exceed that of humans and terrestrial mammals, one-quarter of the specified sample volume was used to bring rT3 into the standard range (0–3000 pM) of the assay. A correction for the altered sample volume was applied after processing. The rT3 assay and the modified procedure have been previously validated (Houser et al., in review). The average replicate CV for rT3 was 3.5%.

### 2.4. Statistical Treatment

Statistical analyses were conducted using JMP statistical software (v.15, SAS Institute Inc., Cary, NC, USA). Variations in circulating hormones were tested using linear mixed models (LMM) with individual dolphin as a random effect and time period as a fixed effect (pre-sample: −1 day before the stress test; the initial, time 0 sample collected immediately after beaching; post-samples collected after TSH injection at: +15, +30, +45, +60 min, +2 and +4 hrs, and +1, +2 and +3 d; see Figure 1). A Dunnett's post-hoc test against the pre-sample (−1 day) was performed if significant differences between post-injection periods were found. As each individual was equally represented at all sample points, linear regression analyses were also performed to explore dynamic relationships between free and bound hormones.

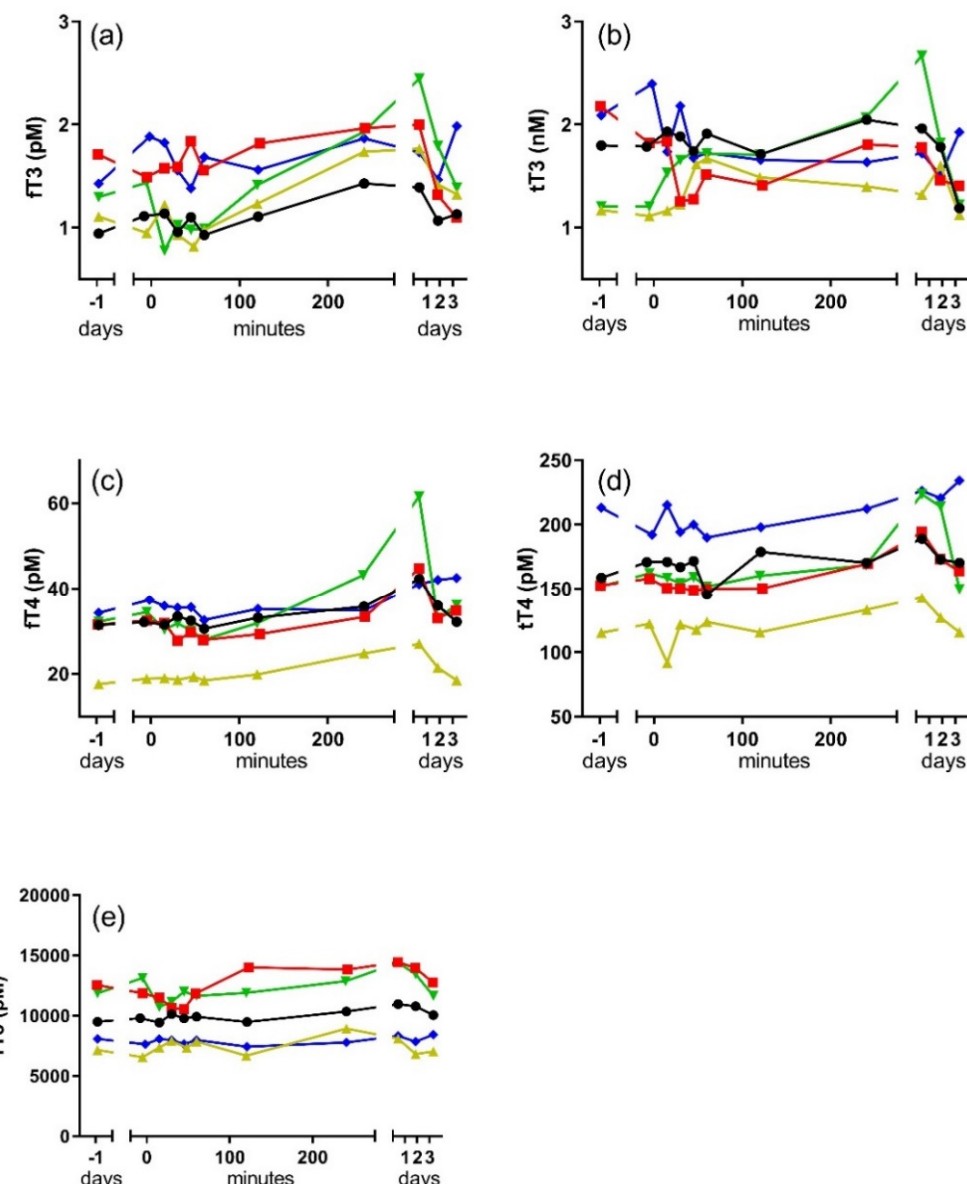

**Figure 1.** Change in bottlenose dolphin thyroid hormones in response to an intramuscular TSH injection. Values along the x-axes are sample times relative to the time of the TSH injection (time = 0). (**a**) fT3, (**b**) tT3, (**c**) fT4, (**d**) tT4, (**e**) rT3.

## 3. Results

Changes in free and bound thyroid hormone in response to the TS are shown for each individual dolphin in Figure 1. Individual levels of circulating thyroid hormones were variable between individuals, but all individuals showed an increase in fT3, fT4, tT4, and rT3 following the TS. Levels of fT3 were significantly higher than controls (−1 day) in blood samples collected four hours and one day after the TSH injection ($F_{10, 40} = 4.0$, $p < 0.001$; Dunnett's $p < 0.05$; Figure 2). Similarly, rT3, fT4, and tT4 were significantly elevated one day after the injection, with tT4 remaining significantly elevated into the second day after the injection (rT3: $F_{10, 40} = 3.0$, $p < 0.01$; Dunnett's $p < 0.05$/fT4: $F_{10, 40} = 7.6$, $p < 0.001$; Dunnett's $p < 0.05$/tT4: $F_{10, 40} = 5.9$, $p < 0.001$; Dunnett's $p < 0.05$). All hormones showing a significant response to the TS returned to baseline by the third day after the TSH injection. Levels of tT3 did not significantly change in response to the TS, nor did ratios of the free and bound forms (fT3:fT4 and tT3:tT4) of the hormone.

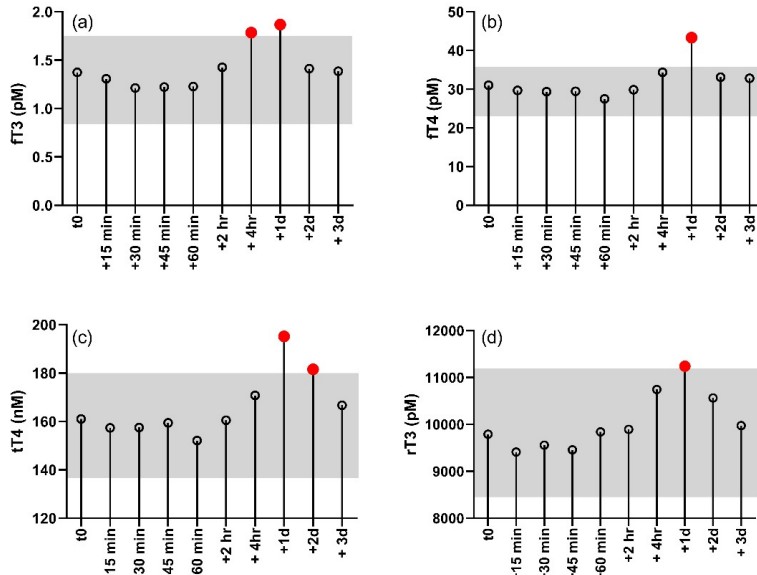

**Figure 2.** Control decision chart displaying changes in the mean value of thyroid hormones relative to the time of the intramuscular TSH injection (t0). The gray shaded box shows the decision limits as determined by Dunnett's test. Red symbols above the shaded box designate measurements that were significantly different from the baseline (determined from the pre-samples taken one day prior to the TS). (**a**) fT3, (**b**) fT4, (**c**) tT4, (**d**) rT3.

Results of regression analyses between thyroid hormones are shown in Figure 3. There was a loose yet significant and direct relationship between fT3 and tT3 ($p < 0.01$, $r^2 = 0.12$). A similar relationship existed between fT4 and tT4, but the amount of variance explained by the linear model was much higher ($p < 0.001$, $r^2 = 0.69$) than that observed between fT3 and tT3. A significant, direct relationship was also observed between fT3 and fT4 ($p < 0.001$, $r^2 = 0.32$). No relationship between rT3 and either tT3 or tT4 was observed, but there were loosely significant relationships between rT3 and fT3 ($p < 0.05$, $r^2 = 0.07$), as well as between rT3 and fT4 ($p < 0.001$, $r^2 = 0.23$).

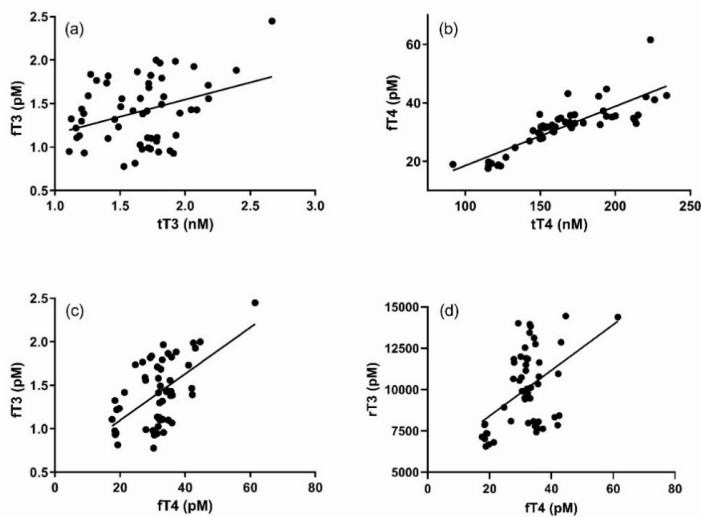

**Figure 3.** Plots of significant direct relationships between (**a**) fT3 and tT3 ($p < 0.01$, $r^2 = 0.12$), (**b**) fT4 and tT4 ($p < 0.001$, $r^2 = 0.69$), (**c**) fT3 and fT4 ($p < 0.001$, $r^2 = 0.32$), and (**d**) rT3 and fT4 ($p < 0.001$, $r^2 = 0.23$).

## 4. Discussion

Stimulation of the thyroid is a potential tool for evaluating thyroid dysfunction. Prior to the development of sensitive TSH assays, the use of TRH was used to differentiate secondary and tertiary hypothyroidism. The use of TSH can similarly be used to directly assess dysfunction in the thyroid gland. TRH stimulation has only been reported once in a toothed whale, the beluga, and produced no significant changes in circulating thyroid hormones [11]. In contrast, the use of both bovine TSH and human recombinant TSH have been used to effectively stimulate the production and secretion of thyroid hormones in belugas and bottlenose dolphins [1,7,8].

West and colleagues [7] were the first to report on the use of human recombinant TSH to stimulate the thyroid of a bottlenose dolphin. A single animal was used in the study and the intramuscular administration of 0.9 mg of TSH produced a modest increase in fT3 (51%) and tT3 (29%), but no distinct pattern was observed for either fT4 or tT4. In the current study, the same human recombinant TSH was utilized for thyroid stimulation, but the dose was increased to 1.5 mg per subject in hopes of producing a more pronounced effect. The sample size was also increased to determine whether changes in thyroid hormones were repeatable across animals. Despite the increased dose of TSH, the mean maximum increase in fT3 was 44%, which was not substantially different from that previously measured [7]. However, whereas no distinctive increase in either fT4 or tT4 was observed by West and colleagues [7], an increase of 47% was observed in fT4 with a more modest mean maximum increase in tT4 of 23%. No significant change in tT3 was observed and rT3 increased by a modest 14%, although the change was significant.

All of the thyroid hormones that significantly increased in response to the TS showed peak concentrations the day after the TSH injection. However, a significant increase was observed in fT3 as early as four hours after the TSH injection and a significant increase in tT4 persisted for up to two days after the injection. The pattern is suggestive of a time lag in thyroid hormone dynamics that is potentially reflective of feedback regulation. Thyroid regulation is sensitive to T3:T4, so it is probably not surprising that the strongest direct relationships existed between fT3 and fT4, as well as fT4 and tT4. A post-hoc analysis of both fT3:fT4 and tT3:tT4 showed no significant change in the ratio of the hormones throughout the post-injection observation period. Nevertheless, the delays between the first periods identified as significantly higher than baseline in fT4 (+one day) and tT4 (+one day) relative to the first period of significant increase in fT3 possibly reflects the time required to change peripheral rates of T4 deiodination in response to a growing excess of fT3. The significant relationship between rT3 and fT4, and the corresponding timing in their significant changes, supports an upregulation of T4 deiodination to the rT3 form.

It is feasible that the testing paradigm employed in the current study affected thyroid hormone levels, as the dolphins involved would have mounted a stress response to the out-of-water portion of the procedure during which sequential blood samples were collected. A similar paradigm was previously used to study the stress response in the bottlenose dolphin and the production of corticosteroids increased rapidly after voluntary beaching and the setting of the needle for sequential blood sampling [3]. However, neither free nor total T3 and T4 demonstrated any significant change in response to the stress test [3], suggesting that samples collected in the current study under a similar paradigm were likely not affected by an acute stress response. Under prolonged elevation of serum cortisol, feedback mechanisms might be expected to suppress thyroid hormone production. At elevated levels, cortisol is known to suppress TSH production and inhibit the conversion of T4 to T3 in humans. Similar to humans, significant reductions in fT3 and fT4 have been observed in bottlenose dolphins under pharmacological elevation of circulating cortisol [12], and tT3 has been observed to decline in response to elevated cortisol in some age classes of the northern elephant seal (*Mirounga angustirostris*) when submitted to adrenocorticotropic hormone stimulations [13].

The results of this study provide baseline information on the response of the thyroid to TSH stimulation that can be utilized for clinical investigation of dolphin thyroid

dysfunction; however, the method could be improved. The TSH utilized in the current study and by West and colleagues [7] was human-derived. Both TSH and its receptor are highly derived for specificity across vertebrate species, minimizing cross-reactivity with glycoprotein receptors, and with receptor specificity and bioactivity potentially peaking with the evolution of the mammals [14]. Thus, it is possible that the effectiveness of the human recombinant TSH was limited in its ability to promote production and secretion of thyroid hormone. Currently, no bottlenose dolphin-derived TSH is available for use in thyroid stimulation tests, nor are there commercial antibodies available to facilitate regular assessment of dolphin TSH. Development of specific antibodies for bottlenose dolphin TSH could greatly improve assessment of thyroid dysfunction, as performed in other species e.g., [15]. Similarly, development of a species-specific thyroid hormone secretagogue (TSH) would presumably enhance the response of the thyroid to stimulation and enable more biologically-relevant kinetic analyses of thyroid hormone release and clearance, e.g., see [16]. The development of such tools is costly and would generally be limited to specific research efforts in exotic species, i.e., it would have limited clinical utility because of its limited availability. However, the bottlenose dolphin has high-profile commercial value due to its abundance in marine parks, zoos, and aquaria, and its value could be used to leverage the development of tools that better enable veterinary assessment of thyroid function in this species.

**Author Contributions:** Each of the authors contributed to this article in the following manner: Conceptualization, D.S.H. and C.C.; Data curation, D.S.H. and C.C.; Formal analysis, D.S.H. and C.C.; Funding acquisition, D.S.H. and D.E.C.; Investigation, D.S.H., C.C. and D.E.C.; Methodology, D.S.H. and C.C.; Project administration, D.S.H.; Resources, D.S.H. and D.E.C.; Supervision, D.S.H.; Visualization, D.S.H.; Writing—original draft, D.S.H.; Writing—review & editing, D.S.H., C.C. and D.E.C. All authors have read and agreed to the published version of the manuscript.

**Funding:** This research was funded by the Office of Naval Research, grant numbers N000141110436 and N000141512230.

**Institutional Review Board Statement:** The study was conducted according to the guidelines of the Declaration of Helsinki, and approved by the Institutional Animal Care and Use Committee of the Naval Information Warfare Center Pacific (protocol #94-2011 approved 1 MAR 2011) and the Department of Navy Bureau of Medicine (protocol NRD-702 approved 15 MAR 2011). Procedures followed all applicable U.S. Department of Defense guidelines for the care of laboratory animals.

**Data Availability Statement:** The data presented in this study are available on request from the corresponding author.

**Acknowledgments:** Special thanks are given to E. Jensen and the trainers and veterinary staff of the US Navy Marine Mammal Program for their support in the performance of this study. This is scientific contribution #303 of the National Marine Mammal Foundation.

**Conflicts of Interest:** The authors declare no conflict of interest. The funders had no role in the design of the study; in the collection, analyses, or interpretation of data; in the writing of the manuscript, or in the decision to publish the results.

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
