# Peer review of "Thyroid-Stimulating Hormone Stimulation Tests in the Bottlenose Dolphin (Tursiops truncatus)"

_2673-5636, doi:10.3390/jzbg2020018_

Round 1

Reviewer 1 Report

Overall, very clean and precise paper with a great amount of background on thyroid function and testing in marine mammals. This paper clearly demonstrates that TSH stimulating testing produced little data that was of clinical value.  However, I am curious if any of the changes observed in thyroid values could be useful to the clinician? If not, it should be clearly stated why the increases in hormone levels and the temporal affects are not useful. .  

I believe the authors need to address the description of physiologic response to stress on line 214-215.  The authors did a good job of describing how increases in cortisone can impact thyroid hormones, however, I don't understand the reasoning for the authors calling the response: so-called "stress response".  

The acute physiologic response to stress (stress response) is a normal biological phenomenon that is beneficial to survival in mammals and is acceptable medical term.

Having a stress response isn't necessarily a negative or invalidates data. Unfortunately, veterinarians and researchers must always take in consideration that physiologic parameters are impacted by stressors when collecting samples or measurements.

I highly recommend the publication of this important data on dolphin thyroid function.

Reviewer 2 Report

This paper is very well written.  Little or no grammatical errors and well edited.  Nice to read.

I have a few mostly procedural/clarification points. Perhaps a few places some more background could be included. 

Lines 56 or so:  Any hypotheses/discussion about why these differences between humans and cetaceans?:  rT3 so much higher and tT4 also higher in comparison to humans?  Interesting.

Isn’t there a comma after e.g., (line 59 and throughout??) 

Line 91:  voluntary “BLOOD” samples

The sampling protocol is a bit unclear. But seem like this:

1 day before

Day of TSH test: 

Pre-injection blood sample

Then 15 minutes for 2 hours post-injection (presumably 8 samples)

2 hours after release

4 hours after release

3 more days post test.

There is confusion with this and results, see below.

There were two different sampling locations?  The bold (day before, and after the 2 hour test period) were from the arteriovenous plexus, and the samples pre-test and during the 2 hours were from the peduncle vessel?????  This is a bit confusing.

Line 123:   Not sure what this means:  ????  One-quarter of the specified sample volume was used to bring the unknown 123 percent bound values into the standard range (0–3000 pM) and a correction for the altered 124 sample volume was applied after processing;  

Line 133:  Sample designation:  You only list 4 samples at 15-minute intervals post injection (15,30,45 and 60), yet above in the design, you say you sampled at 15 minute intervals for 2 hours (assuming then, there should be 75, 90, 105 and 120 minute samples).  You also list only +1 and +2 days post study day, but above in methods you say you sampled for THREE days post TSH injection.  These discrepancies need to be resolved. 

Figure 2:  Not really understanding the “decision limit” graph showing the significance increases???  It is a cool graph showing distinctly which ones are higher than baseline, but I’ve never seen such a graph before.  Nice, but I have no idea of how it works. 

Discussion:  DOSAGE? You doubled here…what is the justification?  Are there other studies of TSH from other animals of similar size?  How does that dosage (per kg body weight) compare to what is useful in a human TSH study???  You imply that dosage may be a significant factor in not getting results…some better context of your dosage decision, and how it relates to other studies, would be interesting.

Paragraph starting line 211: Hmmm, your comments about stress/cortisol are interesting.  It might indeed partially explain the reduced levels early, and significant increases not until the next day, when they are “free”.   So, it brings up this question:  If the animals come for these blood draws relatively easily (day before, hours after, and days after), why do you need to hold them for the 1 hour (not 2 that you talk about)…which as you say, is obviously more stressful for them?  Why can’t you inject, then ask them to come back at 15 minute intervals?  Is that more stressful than staying beached for 1-2 hours??

Also, it is unfortunate that you did not do some cortisol assays along with the thyroid suite here, to spot check these concerns.

Paragraph Line 227:  Just how variable is TSH/receptor across species?  Maybe a bit more detail, besides just the reference, would be useful here?
